# OpenReview forum: "R3HF: Reward Redistribution for Enhancing Reinforcement Learning from Human Feedback"
_ICLR.cc/2025/Conference — ICLR 2025 Conference Withdrawn Submission_

### Official Review · Reviewer_bqTD · 2024-10-16

**Soundness:** 1
**Presentation:** 2
**Contribution:** 1
**Rating:** 1
**Confidence:** 5

**Summary:**

The paper aims to improve the standard RLHF setting for LLMs (optimizing a reward function learned by a Bradley Terry model under a KL constraint) by considering more fine-grained rewards. For this, they seem to essentially propose what is already known in the literature as potential based reward shaping. They choose the potential to be the reward model logit score of intermediate tokens (instead of the full sequence). The paper also contains limited experiments against a small number of baselines.

**Strengths:**

1. The paper considers a relevant topic: how to best align LLMs to human preferences.

**Weaknesses:**

1. The paper seems to be of very limited novelty, as the proposed reward formulation seems to essentially correspond to what is known as potential based reward shaping (see [A]). As a result, the actual contribution of the paper seems trivial: It seems to be limited to the choice of the potential function in the form of the reward model logits of intermediate tokens.
2. In addition, the authors do not clearly motivate why the logit scores of intermediate tokens under the reward model are a good potential function for reward shaping. After all, there is a clear train-test mismatch: SotA reward models are usually only trained by looking at the logits of the last token in the sequence (e.g. an end-of-string-token). It is without motivation why any other logit would yield a reasonable assessment of the quality of the subsequence.
3. As the authors correctly point out, the optimal solution to the (regularized) MDP with potential based reward shaping is the same as the optimal solution to the (regularized) MDP with sequence-level results. This limits the potential of this approach, as we can only hope to get better optimization during finetuning but not a better optimum. The authors do not seem to provide a convincing, principled case why their approach should actually provide better training dynamics.
4. The experimental section is extremely weak as the authors only compare their approach to PPO and DPO. Given that this paper is about improving RL training dynamics (see 3), it should compare to both simple baselines (in particular REINFORCE variants with various obvious choices of sequence and token-level learned and non-learned baselines) as well as SotA RL finetuning algorithms like the inner optimization loop of WARP (see [B]) or BOND (see [C]). In addition, it is well known that hyper parameter tuning is critical for RLHF and there are for example clear interactions between learning rate, regularization, number of steps and final performance, so the authors should add experiments showcasing that their results are robust to the choice of such hyperparameters.

[A] Wiewiora, E. (2003). Potential-based shaping and Q-value initialization are equivalent. Journal of Artificial Intelligence Research, 19, 205–208.
[B] WARP: On the Benefits of Weight Averaged Rewarded Policies https://arxiv.org/abs/2406.16768
[C] BOND: Aligning LLMs with Best-of-N Distillation https://arxiv.org/html/2407.14622v1

**Questions:**

Unless I strongly misunderstood the paper, this paper requires major edits that cannot be addressed in a rebuttal.

---

### Official Review · Reviewer_Hqh7 · 2024-10-26

**Soundness:** 2
**Presentation:** 2
**Contribution:** 1
**Rating:** 3
**Confidence:** 4

**Summary:**

To address the sparse issue in current RLHF, this paper proposes a novel reward redistribution method, R3HF, that allows token-level reward allocation.
Specifically, the proposed method formulates  the reward prediction task as a regression problem that computes redistributed rewards by evaluating the specific contribution of each token to the reward model’s output, which improves the model’s understanding of language nuances.
Experimental results verifies the proposed method.

**Strengths:**

1. The overall direction of learning a dense reward function for RLHF in LMs is promising.
2. The experiments and ablation studies are comprehensive.

**Weaknesses:**

1. Methods for learning a dense per-step reward without additional fine-grained label data has been clearly proposed in the literature, e.g., [1,2] and the reference therein. The authors ought to have a adequate citation, discussion, and ideally comparison with these prior works. Otherwise, the contribution of this work will be significantly unjustifiable. For example, the central idea in L52-85 has been stated similarly in [2].

[1] Feng, Yihao, et al. "Fantastic Rewards and How to Tame Them: A Case Study on Reward Learning for Task-oriented Dialogue Systems." The Eleventh International Conference on Learning Representations.

[2] Yang, Shentao, et al. "Preference-grounded token-level guidance for language model fine-tuning." Advances in Neural Information Processing Systems 36 (2023).

2. Source code seems not anonymously available for review purpose.

3. Eq. (7) seems confusing. IMHO, starting from the second line, all $\tilde r^{RM}$ should be $\mathcal{R}_\phi$.

**Questions:**

1. How is the per-step reward function $\mathcal{R}_\phi$ trained?
2. What is the meaning of "sequence-wide return" and "state-action sequence" in L96?
3. What is the meaning of $p^*$ in L155?
4. In Table 1 (a), which reward model is used to calculate the Average Score? And why is this evaluation reasonable given that the learned reward model may be far from perfect?

---

### Official Review · Reviewer_8UNW · 2024-11-02

**Soundness:** 2
**Presentation:** 2
**Contribution:** 2
**Rating:** 5
**Confidence:** 4

**Summary:**

This paper proposes R3HF, a reward redistribution approach for RLHF. The authors address limitations in traditional RLHF reward allocation and distribute rewards at the token level instead of assigning a single, delayed reward to the entire output sequence. Experimental results show that this fine-grained approach enhances learning efficiency and reduces dependency on human labeling​.

**Strengths:**

The idea of redistributing rewards at the token level in RLHF is compelling. Traditional RLHF methods usually assign a single, sequence-level reward, but these approaches cannot capture the contributions of individual tokens, thus limiting the model’s optimization efficiency and nuanced understanding of language. This paper introduces a method that provides fine-grained, token-level reward allocation.

The workflow of R3HF is reasonable, it provides immediate, token-specific feedback to make language models more responsive to human preferences.

**Weaknesses:**

This paper does not sufficiently explore existing credit assignment and reward redistribution methods in the context of RLHF.

The proposed method relies heavily on the reward model's accuracy, assuming that the regression model can precisely represent each token's reward value in time-difference calculations. Any inaccuracies in the reward model can directly impact reward redistribution, leading to flawed allocations.

Also this method assumes that each token's contribution can be accurately calculated using a time-difference approach, which may not be feasible for sequence generation tasks. For long sequences, and especially when the model relies on context spanning entire sentences or paragraphs, the method may be inadequate.

**Questions:**

Please notice: In your presentation, please pay attention to the inconsistent use of tenses in the experimental section. And the repetitive use of "Eq. equation" impacts readability.

Authors may consider comparing with:

@article{wu2023fine,
  title={Fine-grained human feedback gives better rewards for language model training},
  author={Wu, Zeqiu and Hu, Yushi and Shi, Weijia and Dziri, Nouha and Suhr, Alane and Ammanabrolu, Prithviraj and Smith, Noah A and Ostendorf, Mari and Hajishirzi, Hannaneh},
  journal={Advances in Neural Information Processing Systems},
  volume={36},
  pages={59008--59033},
  year={2023}
}

@misc{chan2024denserewardfreereinforcement,
      title={Dense Reward for Free in Reinforcement Learning from Human Feedback},
      author={Alex J. Chan and Hao Sun and Samuel Holt and Mihaela van der Schaar},
      year={2024},
      eprint={2402.00782},
      archivePrefix={arXiv},
      primaryClass={cs.LG},
      url={https://arxiv.org/abs/2402.00782},
}

---

### Official Review · Reviewer_GAkt · 2024-11-04

**Soundness:** 3
**Presentation:** 3
**Contribution:** 3
**Rating:** 3
**Confidence:** 4

**Summary:**

The paper introduces a novel reward redistribution method known as R3HF, designed to enable a more precise, token-level allocation of rewards. Our approach reframes the reward prediction task of the reward model as a regression problem. Consequently, the redistributed rewards are determined by assessing the individual contribution of each token to the output of the reward model.

**Strengths:**

* The paper proposes a novel reward shaping method in the LLM setting.
* The experiment result is good.

**Weaknesses:**

* The reward design seems heuristic, and it is probably not fit for some complex reasoning settings.
* I don't see too much advantage from Figure 3.

**Questions:**

* Can this method scale to more datasets like reasoning tasks (math, coding, etc.)?
* Can you analyze the reward distribution of your method?

---

### Official Review · Reviewer_iha9 · 2024-11-04

**Soundness:** 1
**Presentation:** 2
**Contribution:** 1
**Rating:** 3
**Confidence:** 4

**Summary:**

This work proposes a modification to reward labelling for RLHF training. Instead of labelling each sequence with a single reward at the EOS token, "reward redistribution" proposes to give a reward to each token (like a value function). The reward for a token is the difference in reward model score if this token is added. The reward model is not re-trained but used as is for this, but this changes the loss for PPO training.

They demonstrate their method applied with PPO outperforms a baseline PPO on question-answering, a modified TLDR, and safeRLHF. They evaluate win-rates against their own trained SFT model using their own reward model as well as GPT-4.

**Strengths:**

The paper is easy to read and describes the method succintly.

**Weaknesses:**

This work is fundamentally flawed theoretically, practically, and in experiment setup. I believe it is impossible for the authors to adequately addresss my concerns in the rebuttal period and recommend they fundamentally rethink their direction.

Theoretically, RLHF is not a sequential MDP as said by the authors and this invalidates the need for reward redistribution. The authors say that generating N tokens from an LLM can be viewed as N steps in an MDP. But this assumes that: 1. states of the MDP are simply the current sequence 2. Taking an action is equivalent to choosing the next token But 3. there is no transition probability because choosing a next token deterministically adds the token to the current sequence. Single-turn RLHF is more accurately seen as a single step: generating a full sequence of tokens and then getting feedback from a reward model. This is not an MDP but a contextual bandit. If generating a full sequence is a contextual bandit then it makes sense for it to have a single reward, and not need N rewards per token. Their approach could be shown as empirically effective, but it isn't.


Practically, it has been shown on TLDR that the learned reward model is not an effective method of labelling individual tokens with reward (Huang et al, 2024). Trained reward models are explicitly not good at giving reward for non-EOS tokens. Instead, actor-critic methods explicitly learn a value function that does this instead. The authors somehow miss the clear connection that they are trying to create a value function from the reward model. But this doesn't make sense since their baseline method, PPO, already learns a value function! Furthermore, recent works like RLOO (Ahmadian et al, 2023) have argued that value functions are fundamentally not good in the LLM setting. The authors never compare their method to value functions, to monte-carlo estimates of value (like RLOO), and do not have any experiments explaining the qualitative benefits of their approach and *why* it works in practice.

Experiment setup is poor. The TLDR task they use is not at all equivalent to the original TLDR task (Stiennon et al, 2020) because the SFT dataset is somehow 14,900 examples and not ~117000 examples as it should be. The authors should follow Huang et al (2024) for open-source TLDR experiments. The authors use LLaMA 1 which is out-dated at this point. They also compare against their own SFT baseline as opposed to existing human-written baselines, which is standard practice (again see Huang et al (2024)). RLHF should not be evaluated using point metrics but with curves comparing KL to performance (Gao et al, 2022). Other issues exist but I cannot enumerate them all. Evals should also not use the training reward as this is explicitly trained with (the authors also have GPT-4 win-rates but these results are few). Figure 5 demonstrates clear overfitting of the baselines but not the authors method, simply reducing the learning rate of baselines would probably fix their performance. There are other issues but I cannot enumerate them all.

Overall, the approach is quite weak. The authors would make a much stronger case if they showed their reward redistribution is actually effective on qualitative examples. Furthermore, it makes no sense why the reward model would be effective on non EOS tokens, and perhaps training it to fulfill the requirements of section 3.2 makes more sense.

**Questions:**

What is the point of section 3.4 (connection to DPO) ? DPO assumes a contextual bandit formulation, so your comparison is moot.

Why are your results in Table 2a so different from Table 2b? DPO is the worst-performing method by far in 2a and essentially the same as your top-performing method in 2b. Clearly your reward model and GPT-4 are not aligned.

---

### Note · Authors · 2024-11-12

I have read and agree with the venue's withdrawal policy on behalf of myself and my co-authors.